# High Sensitivity Planar Hall Effect Magnetic Field Gradiometer for Measurements in Millimeter Scale Environments

**DOI:** 10.3390/mi13111898

**Published:** 2022-11-02

**Authors:** Hariharan Nhalil, Moty Schultz, Shai Amrusi, Asaf Grosz, Lior Klein

**Affiliations:** 1Department of Physics, Institute of Nanotechnology and Advanced Materials, Bar-Ilan University, Ramat-Gan 52900, Israel; 2Department of Electrical and Computer Engineering, Ben-Gurion University of the Negev, P.O. Box 653, Beer-Sheva 84105, Israel

**Keywords:** planar hall effect, gradiometer, magnetic sensor

## Abstract

We report a specially designed magnetic field gradiometer based on a single elliptical planar Hall effect (PHE) sensor, which allows measuring magnetic field at nine different positions in a 4 mm length scale. The gradiometer detects magnetic field gradients with equivalent gradient magnetic noises of ∼958, ∼192, ∼51, and ∼26 nT/m√ Hz (pT/mm√Hz) at 0.1, 1, 10, and 50 Hz, respectively. The performance of the gradiometer is tested in ambient conditions by measuring the field gradient induced by electric currents driven in a long straight wire. This gradiometer is expected to be highly useful for the measurement of magnetic field gradients in confined areas for its small footprint, low noise, scalability, simple design, and low costs.

## 1. Introduction

The detection of magnetic field gradients is important for many applications, including geomagnetic mapping [1,2], magnetic navigation [3,4], current sensing [5,6], medical diagnosis [7,8], etc. To perform detections in confined spaces, especially where the compact form factor is highly important (e.g., any portable [9] or wearable electronic devices [10,11,12], body implants [13], etc.), the sensors should have low equivalent magnetic noise and a small footprint as the minimum baseline (the distance between two sensors) of the gradiometer is decided by its footprint. Although there are gradiometers that can quantify the field gradient by measuring the magnetic field only at a single point [14], commonly, the determination of the magnetic field gradient requires magnetic field measurements at two or more locations.

Planer Hall effect sensors have the features required for confined space gradiometry: small footprint and high field resolution. Compared to other magnetoresistive sensors, they exhibit very low equivalent magnetic noise (EMN or resolution), especially at very low frequencies [15,16,17]. Previously, EMNs of ∼5 pT/√Hz at 10 Hz and less than 25 pT/√Hz at 50 Hz were reported for elliptical PHE sensors with and without magnetic flux concentrators (MFCs), respectively [16,17]. Here, the area of the magnetic ellipse with a major axis length of 5 mm (aspect ratio of 1/6) without MFCs is only ∼5 mm${}^{2}$, whereas for high-resolution sensors such as superconducting quantum interference device (SQUID) magnetometers [18] or conventional fluxgate magnetometers [19], they have at least a one-order higher footprint, which makes them unsuitable for many confined space measurements. Moreover, some of these high-resolution magnetometers are operated at cryogenic temperatures and require bulky instrumentation. PHE sensors on the other hand are known for their room temperature operation, low temperature dependence [20], simple design and fabrication, linearity, and portability. These important characteristics make them ideal for gradiometer applications in confined spaces.

Here, we discuss the fabrication and testing of a specially designed elliptical planar Hall effect gradiometer (EPHEG) with nine sensing positions in a 4 mm length scale and a footprint less than 5 mm${}^{2}$. Since the excitation current is shared by all sensing positions, the power consumption is 9 times smaller than the power needed to operate nine separate sensors. Moreover, the EPHEG can be used in smaller volumes than nine separate elliptical sensors and has redundancy in case one voltage pair fails during a measurement. Our EPHEG presents low equivalent gradient magnetic noises of ∼958, ∼192, ∼51, and ∼26 nT/m√Hz at 0.1, 1, 10, and 50 Hz. We also demonstrate the gradiometer performance by measuring the field gradient induced from a current-carrying long straight wire in ambient (outside the magnetic shield) conditions.

## 2. Experimental Techniques

Al${}_{2}$O${}_{3}$(60 nm)/Ta(5 nm)/Permalloy (Ni${}_{80}$Fe${}_{20}$, Py)(50 nm)/Ta(5 nm) films are sputter deposited on Si/SiO${}_{2}$(naturally oxidized) wafers by ion beam sputtering (Intelvac Nanoquest I, Ontario, Canada). Elliptical shapes with a major axis (*a*) length of 5 mm and minor axis (*b*) length of 0.833 mm are patterned via photo-lithography, ion milling, and wet etch processes. Gold electrical contact pads of ∼1.5 times the thickness of Py are deposited and patterned via photo-lithography and lift-off processes in a second stage. Figure 1 shows a schematic diagram of an EPHEG including the configuration and dimensions of its electrical contacts. Sensitivity is measured in zero dc field by measuring the PHE signal with an ac magnetic field (100–200 nT) generated by a home-built, calibrated solenoid kept inside a magnetic shield (AMUNEAL MFG CORP, Philadelphia, PA, USA). For noise characterization and sensitivity measurements, the sensor is excited along the magnetic easy axis (long axis of the ellipse) with an ac current (1210 Hz) of different amplitudes employing a PXI-5421 function generator. Using an ultra-low noise transformer-matched amplifier (TMA, home built) [21], the transverse PHE voltage is amplified and digitized using a 24-bit sigma-delta analog to digital converter model PXIe 4464, from National Instruments. Noise measurements are performed inside a 3-layer Mu-metal magnetic shield to mitigate the effect of the ambient field and electromagnetic interferences. Gradient sensitivity and equivalent gradient magnetic noise are calculated from the measured sensitivity and voltage noise spectral densities of different voltage pairs, respectively. For testing and demonstration, the magnetic field from a long thin straight wire (length ∼1.5 m, radius ∼0.1 mm) carrying an ac current (5 Hz) is measured using the same instruments and electronics used for noise measurements in an open environment (outside the magnetic shield) and compared to the theoretical values calculated by Ampere’s law. All measurements are performed at room temperature.

## 3. Equivalent Magnetic Noise Characterizations

Our EPHEG consists of nine parallel voltage pairs that are equally spaced along the x-axis with a separation of 0.5 mm between each pair (Figure 1). To explore the position dependence of EMN, we measured the sensitivity ($S_{y}$) and noise at all voltage pairs.

The $S_{y}$ of the EPHEG is defined as the ratio between the transverse voltage $V_{y}$ (voltage across the parallel voltage pairs 1 to 9 in Figure 1) and the in-plane magnetic field *B* applied along the short axis of the ellipse given a current $I_{x}$ flowing between $I_{x1}$ and $I_{x2}$ (Figure 1). For magnetic fields much smaller than the magnetic anisotropy field, $S_{y}$ in V/T is given by [22]:(1)$$S_{y}=\frac{V_{y}}{B}=I_{x}\cdot \frac{\Delta \rho }{t}\cdot \frac{1}{H_{eff}}=I_{x}\cdot \Delta R\cdot \frac{1}{H_{eff}}$$
where Δρ is ($\rho_{\|}-\rho_{⊥}$), $\rho_{\|}$ and $\rho_{⊥}$ are the resistivities when the magnetization is parallel and perpendicular to the current, *t* is the film thickness, ΔR is $\frac{\Delta \rho }{t}$, and $H_{eff}$ is the effective anisotropy field, which is the sum of the shape-induced anisotropy and growth-induced anisotropy [23].

$S_{y}$ for all the nine parallel voltage pairs is measured for several excitation currents and is found to be linear with the excitation (Appendix A). $S_{y}$ for $I_{x}$∼130 mA (peak) and different voltage pairs is listed in Table 1. We note a systematic change in $S_{y}$; $S_{y}$ is maximum for the middle voltage pairs, and it decreases towards edge pairs. Based on Equation (Equation 1), $S_{y}$ depends inversely on $H_{eff}$. $H_{eff}$ is lowest at the middle voltage pair (pair 5) and increases gradually towards the edge pairs. The reduction in $S_{y}$ (∼21% reduction from the middle to edge pair), however, is mostly due to the reduction in ΔR rather than the nominal increase in $H_{eff}$ (only ∼ 6%).

$R_{y}$ (transverse resistance) is at a maximal value at the middle voltage pair and decreases towards the edges due to the increase in the transverse distance ($b_{n}$) between the parallel voltage pairs (see the schematic in Figure 1). $R_{y}$, $H_{eff}$, ΔR, and $b_{n}$ for different voltage pairs are also tabulated in Table 1. The two-probe longitudinal resistance ($R_{x}$) and AMR ratio of the film are ∼38 Ω and ∼1.9 %, respectively.

Equivalent magnetic noise (EMN or $B_{eq}$) is defined as follows [21,22]:(2)$$B_{eq}\left(f\right)=\frac{e_{\Sigma }\left(f\right)}{S_{y}}$$
where $S_{y}$ and $e_{\Sigma }$ are the sensitivity (given by Equation (Equation 1)) and total noise spectral density, respectively. $e_{\Sigma }$ has different contributions from 1/*f*, Johnson (thermal), and preamplifier noises and can be expressed as follows [21]:(3)$$e_{\Sigma }\left(f\right)=\sqrt{I_{x}^{2}R_{x}^{2}\frac{\delta_{H}}{N_{c}^{.}Vol^{.}f^{\alpha }}+4k_{B}TR_{y}+e_{amp}^{2}}$$
where $\delta_{H}$ is the Hooge parameter [24]. $N_{C}$ is Permalloy’s free electron density of (1.7 × 10${}^{29}$/m${}^{3}$). Vol is the sensor’s effective volume, where the electrons contribute to the conduction process in a homogeneous sample. *f* is the frequency. α is a constant. $k_{B}$ is the Boltzmann constant. *T* is the temperature of the sensor. $R_{y}$ is the resistance across the voltage terminals, and $e_{amp}$ is the total noise of the TMA [21]. $B_{eq}$ is obtained by measuring $S_{y}$ and $e_{\Sigma }$ separately [22].

Each EMN vs. frequency data between 0.1 and 100 Hz is fitted with Beq(f)=c02+(c1fβ)(2 (where $c_{0}$, $c_{1}$, and β are fit parameters [17]). From the fit, EMN at a specific frequency is extracted. EMN for different voltage pairs with an excitation current of ∼130 mA is shown in Figure 2 for four different frequencies. It can be seen that the best EMNs are in the middle voltage pairs and can be attributed to the observed high sensitivities and comparatively larger effective volumes (due to larger $b_{n}$). Similar changes in EMN with voltage-pair positions are also seen for lower excitation (Appendix A). The best EMNs measured are ∼29 pT/√Hz, ∼62 pT/√Hz, ∼279 pT/√Hz, and ∼1322 pT/√Hz at 50, 10, 1, and 0.1 Hz, respectively, for voltage pair number 5. The obtained EMNs closely resemble those in our previous study with respect to similar thickness (50 nm) elliptical PHE sensors where we explored the thickness dependence of EMNs [17].

## 4. Equivalent Gradient Magnetic Noise

Our EPHEG with nine voltage pairs can be treated as an array of nine sensors that are placed linearly at 0.5 mm separation between each pair. Considering that the noise measurements are performed in a three-layer magnetically shielded environment, and by employing extremely low noise TMA, the noise sources of any two voltage pairs can be approximated to be completely uncorrelated. The equivalent gradient magnetic noise (gradient resolution, EGMN, or $grad\_B_{eq}$) of two sensors with noise sources that are uncorrelated can be defined as follows.
(4)$$grad\_B_{eq}=\frac{Total\;voltage\;spectral\;noise\;density(f)}{Gradient\;sensitivity}$$

Here, the total voltage spectral noise density (TVSND) can be calculated from individual voltage spectral noise densities (VSNDs) of the two separate sensors as follows:(5)$$TVSND\left(f\right)=\sqrt{{VSND_{1}}^{2}\left(f\right)+{VSND_{2}}^{2}\left(f\right)}$$
where $VSND_{1}$ and $VSND_{2}$ are the voltage spectral noise densities of sensors 1 and 2, respectively, in $V/\sqrt{\mathrm{Hz}}$. The denominator gradient sensitivity (GS) in Equation (Equation 4) is the sensitivity towards the gradient field and is defined as follows:(6)$$GS=|\frac{B_{1}\cdot S_{y1}-B_{2}\cdot S_{y2}}{gradB}|$$
where $B_{1}$ and $B_{2}$ are the magnetic field acting on sensors 1 and 2, and $S_{y1}$ and $S_{y2}$ are their respective sensitivities in V/T. gradB is the magnetic field gradient in T/m.

Equation (Equation 6) can be rewritten as follows:(7)$$GS=|\Delta d\cdot S_{y1}+\frac{B_{2}\left(S_{y1}-S_{y2}\right)}{gradB}|$$
where Δd is the distance between the sensors. Equation (Equation 7) shows that GS depends both on the strength of the magnetic field and the gradient field. GS does not depends on the field strength only when both sensors have the same sensitivity: $S_{y1}=S_{y2}$. Selecting a pair of leads with a larger distance will result in higher gradiometer sensitivity. On the contrary, as this distance increases, the capability of the gradiometer to reject interferences generated by other sources of field decreases. This is an important parameter, for example, in real-world scenarios where many sources of interference are present [25]. The multi-lead approach applied with this sensor allows the user to select an optimal sensitivity/rejection configuration that is most applicable to his specific use case. In the case of EPHEG, for obtaining a better GS, we choose two voltage pairs that are farther apart as possible and have the same or very similar $S_{y}$. From Table 1, it is clear that $S_{y}$ of voltage pairs that are relatively at the same positions from the middle of the sensor have similar sensitivities; i.e., voltage pairs 1–9, 2–8, 3–7, and 4–6 have similar sensitivities. Out of these 4 voltage pairs, 1–9 are the farthest pairs and are expected to provide the highest GS and subsequently the best EGMN.

From the $S_{y}$ of each voltage pair measured for the same excitation, GS is calculated for two voltage pairs having similar sensitivities for an arbitrary constant ΔB of 1 nT (ΔB = $B_{1}$−$B_{2}$, $B_{1}$ and $B_{2}$ are the magnetic field at positions 1 and 2, respectively). TVSND is calculated from the respective VSNDs according to Equation (Equation 5) and EGMN is calculated. The calculated frequency dependence of EGMN is fitted with the following:(8)grad_Beq(f)=a02+(a1fγ)(2
where $a_{0}$, $a_{1}$, and γ are fit parameters. Figure 3 shows the EGMN vs. frequency for four different dual voltage pairs. For clarity, data and fit together are shown only for the 1–9 dual voltage pairs, and for all others, only the fit according to Equation (Equation 8) is shown. As expected, the best EGMN is obtained when measurement voltage pairs are the farthest apart. The EGMN obtained are ∼958, ∼192, ∼51, ∼26, and ∼25 nT/m√Hz (pT/mm √Hz) at 0.1, 1, 10, 50, and 100 Hz, respectively, with fit parameters $a_{0}$, $a_{1}$, and γ as 3.76 × 10${}^{-9}$, 4.43 × 10${}^{-8}$, and 0.69, respectively.

In comparison with other gradiometers, SQUID-based gradiometers have the best reported gradient field resolution <100 fT/m√Hz followed by atomic magnetometers with <10 pT/m√Hz [26,27,28]. Fluxgate and magnetoelectric gradiometers reported gradient resolutions <1 nT/m√Hz [29,30,31]. If we compare the EGMN of a fluxgate gradiometer with baselines of 100 mm [29] and 600 mm [30], they have 0.52 nT/m/$\sqrt{\mathrm{Hz}}$ and 0.03 nT/m/$\sqrt{\mathrm{Hz}}$ gradient resolutions, respectively, and they are at least two orders higher than our EPHEG’s EGMN. However, the reported baselines of these sensors are several times higher than that of our EPHEG sensor, which has a maximum baseline of 4 mm and a minimum baseline of 0.5 mm. The EMN of our EPHEG ‘s middle pair is 29 pT/$\sqrt{\mathrm{Hz}}$, which is better than or within the range of many high-end MEMS fluxgate sensors of smaller size [32,33]. However, the very small baseline of EPHEG has a negative impact on the overall EGMN.

To demonstrate the performance of the EPHEG, the field gradient from an ac current (1 mA, 5 Hz) carrying a long thin straight wire (length ∼1.5 m) is measured in the ambient conditions (outside the magnetic shield) and compared with the theoretical values. For measurements, a calibrated EPHEG is placed near the wire such that the field lines from the wire are in-plane and perpendicular to the long axis of the sensor, as shown schematically in the inset of Figure 4. Starting from a position very close to the wire, the EPHEG is moved (in 4 mm steps) radially outwards on a rail that has precision control on the lateral movements. At each position, the magnetic field is measured by the five different voltage pairs (five alternate pairs covering a total distance of 4 mm) separately with a bandwidth of 0.1 Hz. After measuring all five pairs at a position, the sensor is moved radially outwards 4 mm to a new position and all five pairs are measured again. This procedure is repeated to obtain the field vs. distance plot shown in Figure 4. The error bar on each measurement point represents the standard deviation of ten measurements, and the experimentally measured magnetic fields agree well with the theoretical field values. The lowest gradient field calculated from this measurement is ∼100 nT/m at ∼40 mm.

The gradient field resolution improves with the increase in the baseline, and in applications where the gradient field is relatively homogeneous in the sensor’s volume, measurement errors can be reduced by averaging the output of several sensor pairs [34,35]. To find out the performance limit of EPHEG, the magnetic field is measured by five different voltage pairs (averaging 20 measurements at each voltage pair) that are separated 1 mm apart (i.e., alternate voltage pairs) covering a total length of 4 mm at a distance of ∼ 40 mm from the wire. At this distance, the magnetic field variation with radial distance from the wire can be approximated to be linear and the slope will give the field gradient. Figure 5 shows the magnetic field from a wire carrying 0.2 mA and 0.4 mA measured along with linear fits. We define the limit of the measurement when the slope of the experimental curve deviates ±10% from the slope of the theoretical one. Accordingly, the lowest magnetic field gradient measured is ∼20 nT/m (slope = −0.0254 nT/mm) for 0.2 mA and ∼40 nT/m (slope = −0.045 nT/mm) for 0.4 mA. In terms of linear fit errors, measuring five pairs has a nominal advantage over measuring two pairs. These values are higher than the EGMN determined for two voltage pairs inside the magnetic shield and the difference could be attributed to the harsh measurement environment. Below 0.2 mA, even though the sensor was able to sense the magnetic field, the gradient field calculation is erroneous, and for excitations below 0.1 mA, the signal was below the noise floor. Lower bandwidth measurements utilizing all nine voltage pairs are expected to improve these results.

A comparison of gradient field detectivity of several different types of gradiometers by converting a uniform field sensor to a gradiometer of baseline of l cm is reported by J. Javor et al. [14]. Notably, the detectivity of the EPHEG is within the range of the microelectromechanical system (MEMS) [14], Hall effect [36], Fluxgate [19,37], and the majority of magnetoresistive gradiometers [14]. Since our EPHEG can sense the magnetic field at different positions simultaneously, it can find applications in bio-magnetic sensing [38,39], drug delivery [40,41], and other fields where the speed and flow rate of the bio-magnetic particles need to be determined.

## 5. Summary

In summary, we report the fabrication and performance of a permalloy-based elliptical planar Hall effect magnetic field gradiometer with nine field-sensing positions. Using two field sensing positions with a baseline of 4 mm, it can measure the magnetic field gradients with equivalent gradient magnetic noises of ∼958, ∼192, ∼51, and ∼26 nT/m√Hz (pT/mm√Hz) at 0.1, 1, 10, and 50 Hz, respectively. We demonstrate the performance of the EPHEG by measuring the field gradient from a current-carrying wire using five voltage pairs with a minimum detectivity of ∼20 nT/m in ambient conditions outside the magnetic shield. Our simple, low-cost, high-resolution, and small-footprint gradiometer is expected to be highly useful in measuring gradient fields in confined spaces.

## Figures and Tables

**Figure 1 micromachines-13-01898-f001:**
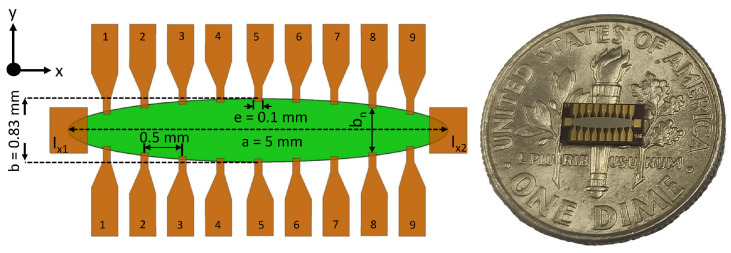
(**left**) Schematics of EPHEG geometry (not to scale). The orange regions are gold electrical contact pads. The long and short axes of the ellipse are labeled as *a* and *b*, respectively. The sensor is excited between I${}_{x1}$ and I${}_{x2}$, and the transverse signal is measured across any of the parallel 9 voltage pairs (marked from 1 to 9). $b_{n}$ is the distance between voltage pairs *n*. (**right**) Photograph of a sensor on a dime as a size reference.

**Figure 2 micromachines-13-01898-f002:**
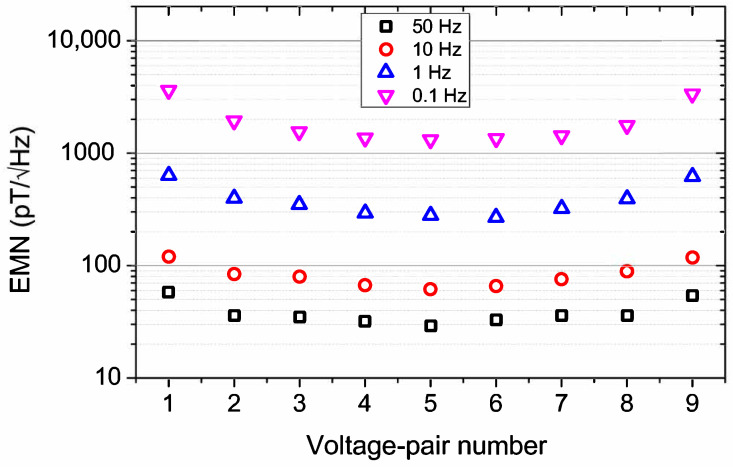
Plot showing the variation of EMN with the position of the voltage pairs for 50, 10, 1, and 0.1 Hz frequencies and for the highest excitation current of ∼130 mA.

**Figure 3 micromachines-13-01898-f003:**
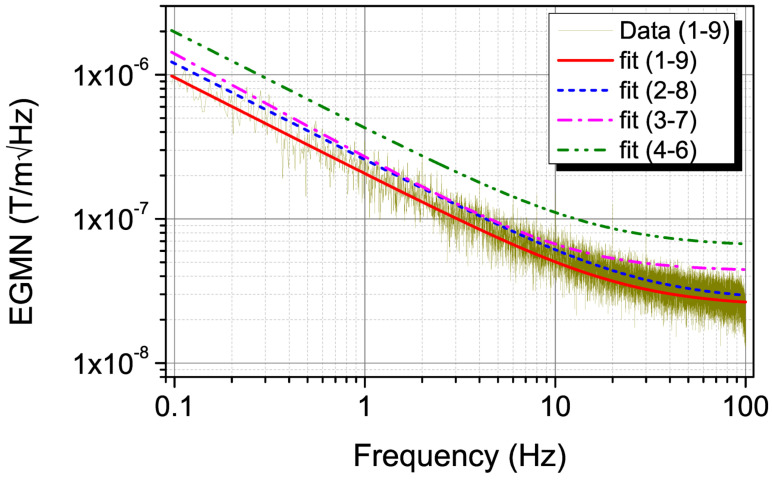
Frequency– dependence of EGMN of 4 voltage pairs 1–9, 2–8, 3–7 and 4–6. Data and fit according to Equation (Equation 8) are shown only for the 1-9 voltage pairs. For clarity, only the fit is shown for remaining pairs.

**Figure 4 micromachines-13-01898-f004:**
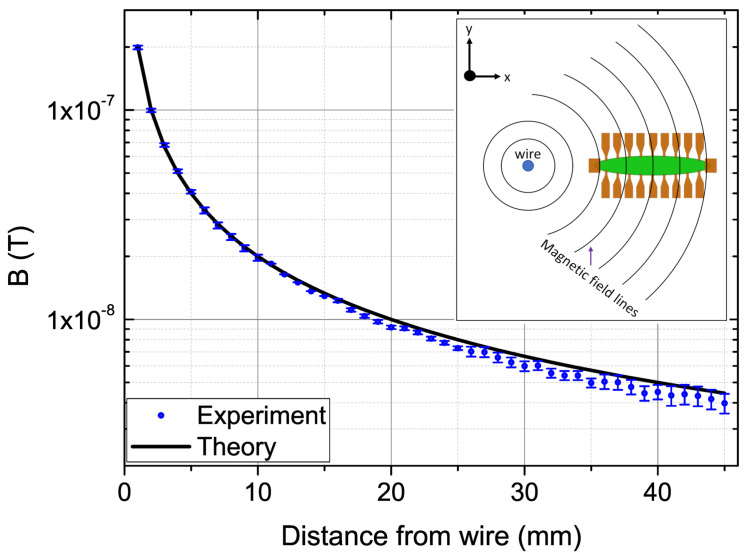
The variation of the magnetic field from a current-carrying wire with a distance from the wire (measured outside the magnetic shield) with an EPHEG along with the calculated magnetic field. The standard deviation of 10 measurements is presented as the error at each point. The inset depicts the schematics of the measurement setup. The current-carrying wire is in the z-direction perpendicular to the plane of the figure.

**Figure 5 micromachines-13-01898-f005:**
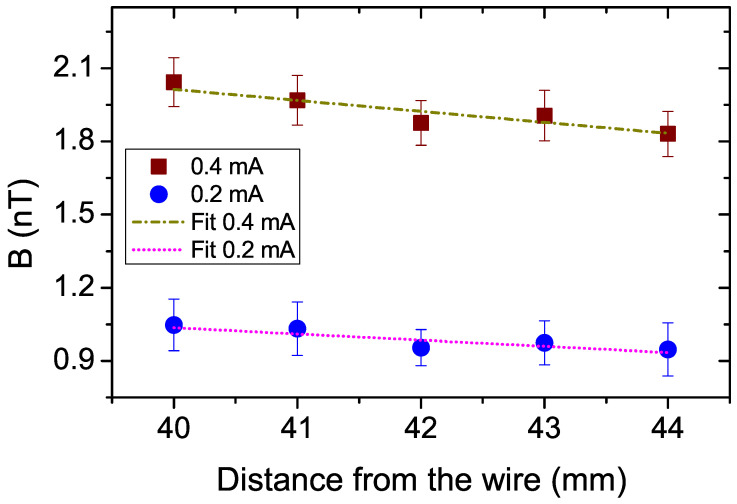
The magnetic field from a wire carrying 0.2 mA and 0.4 mA measured at a distance of ∼40 mm using five voltage pairs. The error bar shown for each data point represents the standard deviation of 20 measurements.

**Table 1 micromachines-13-01898-t001:** Typical values of $R_{y}$, $H_{eff}$, $S_{y}$ (for $I_{x}$ = ∼130 mA (peak)), ΔR, and $b_{n}$ for different parallel voltage-pairs of EPHEG.

V-pair	$R_{y}$	$H_{\mathrm{eff}}$	$S_{y}$	ΔR	$b_{n}$
No:	(Ω)	(Oe)	(V/T)	(Ω)	(m)
1	10.05	5.68	17.83	0.0778	4.200 × 10${}^{-4}$
2	11.14	5.44	22.43	0.0938	5.866 × 10${}^{-4}$
3	11.53	5.43	23.06	0.0964	6.837 × 10${}^{-4}$
4	11.57	5.35	23.62	0.0973	7.365 × 10${}^{-4}$
5	11.79	5.35	23.68	0.0975	7.533 × 10${}^{-4}$
6	11.61	5.43	23.39	0.0974	7.365 × 10${}^{-4}$
7	11.49	5.46	23.02	0.0966	6.837 × 10${}^{-4}$
8	11.07	5.52	22.70	0.0966	5.866 × 10${}^{-4}$
9	10.12	5.66	18.01	0.0781	4.200 × 10${}^{-4}$

## Data Availability

Not applicable.

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
