# Peer review of "High Sensitivity Planar Hall Effect Magnetic Field Gradiometer for Measurements in Millimeter Scale Environments"

_micromachines, 2022, doi:10.3390/mi13111898_

Round 1

Reviewer 1 Report

The authors have given a wonderful result.

Magnetic field gradiometer is a very important instrument for various applications and suitable for MDPI publication. The authors report a new gradiometer with low magnetic noises, which is a wonderful result. However, some infomation should be added   (1) Page 3, the authors report,"Sy for all the 9 parallel voltage pairs is measured for several excitation currents and is found to be linear with the excitation. ", but no experiment data is shown. We cannot tell the linearity as the authors claim.  

(2) On Page 2, the authors report, "For noise characterization and sensitivity measurements, the sensor is excited with an ac current at 1210 Hz with different amplitudes along the magnetic easy axis (long axis of the ellipse) using a PXI-5421 function generator.", but in paragraph "3. Equivalent magnetic noise characterizations" no discussion for different amplitudes at 1210Hz. The authors should add some experiment data.

(3) Please add some data for explaining "Sy for all the 9 parallel voltage pairs is measured for several excitation currents and is found to be linear with the excitation. "

Reviewer 2 Report

Nhalil et. al. report an improvement on PHE field sensors by utilizing closely spaced voltage pairs for a sensitive gradiometric measurement of a current carrying wire. This is a novel and useful device and will be a good publication in Micromachines if the comments below are addressed appropriately.

The authors report a sensitivity in the units of nT/m and are claiming a novelty in the application of measuring gradients in "small" places with millimeter separations. It would be far more appropriate to report sensitivities with a unit of nT/mm or uT/mm. The reported number of 26 nT/m is equivalent to 26 uT/mm, and would be a far more appropriate value. The unit of nT/m is generally appropriate if one is measuring very low spatial frequencies of magnetic fields, such as that of the Earth's magnetic field.

Furthermore, the authors claim there is a value to confined space gradiometry but do not mention any such applications. They should note at least a few applications where such measurements are important, where today's technology cannot achieve such a measurement. One example that I might think of is in a mobile phone or portable electronic device. Here, compact form factor is highly important, and one can position a small sensor nearly arbitrarily close to a current-carrying wire, which happens to be what you demonstrated! Moreover, gradiometry is typically very important due to electromagnetic interference in ambient environments and also from other devices on the phone. Such a technology might be able to better estimate battery health or monitor/optimize power consumption in mobile technologies.

The authors claim that one can only achieve a gradient magnetic field measurement by measuring the magnetic field intensity at two or more locations (line 13-14). This is not true and has been measured at a "single point" by the force on permanent micromagnets in several cases. You already have cited a publication that does this in #28. Please add a sentence to note this. The equivalent spatial separation of that device is 0.25mm, whereas yours is 0.5mm. 

In section 4, you mention that the set up is measured inside a 3 layer magnetic shield. Please name the product/manufacturer and mention the attenuation in the product's spec sheet. Often, 3 layer shields are near -100dB, but all products differ. You might consider mentioning where the closest EM interference sources are as well, relative to your sensor. Is the x-axis of your sensor parallel to those sources for better cancellation? Which direction is Earth's magnetic field? Do you expect that all these sources are attenuated beneath the noise floor of sensor? This work (or a future one) could be greatly improved by demonstrating the performance of the PHEG in an ambient magnetic environment. 

In this light, please change every place in the text where you say "ambient environment" to "ambient temperature and pressure." It is misleading to say this since all your measurements are in a magnetic shield. 

In Figure 3, it surprising to not see 60Hz interference while in a laboratory environment, even from inside a shield. Please discuss this! Is this a result of excellent cancellation in the 1-9 pair with very similar Sy? In that case, do you see 60Hz interference on the other pairs where Sy does not match as well? Please include a [potentially supplementary] figure with EGMN vs frequency for the other pairs so that this is clear. This information cannot come from just looking at the fit.

Please discuss Eq 8, where it came from, and why it is appropriate for this fit. 

In line 92, you have grossly mis-reported the sensitivity of SQUID and atomic magnetometer technologies. You write 100fT/m for SQUID and 1 pT/m for atomic magnetometers. Even your own citation (#21) reports in the title that an atomic magnetometer was developed to measure fields smaller than 20 fT/cm/rtHz. In your units, this would be 0.2 fT/m/rtHz. This means you were off by a factor of 5000 in your text. Please replace these citations with relevant ones that reflect the state of the art in SQUID and AM technology.

It also appears that you have misrepresented fluxgate technology, as one of your own citations (#24) reports 30 pT/m/rtHz. Please correct this.

Also, the last sentence of this paragraph (line 96) could be argued more effectively. I suggest you compare the equivalent magnetic noise of one of your voltage pairs to the state of the art fluxgate. Then, you may discuss that your measurement points are far closer than theirs, affecting the equivalent gradient resolution. 

In line 105, what are the "experimental constraints" that limit you to measuring only 5 voltage pairs? You might just delete this as it does not really matter to the reader. You have also previously stated that you only need two voltage pairs for a gradient measurement...

So you mention that you record 5 voltage pairs for Fig 4, but how are you calculating the y axis values? Is each point an average of all 5 pairs? If so, you need to mention this in the text and in the figure caption. Otherwise, you are leading the reader to believe that your technology is far greater resolution than reality. Additionally, the center point might be varying slightly if you are averaging different positions, contributing to greater error. Also, this is a bit inconsistent with most of your text, which discusses measurement with one or two voltage pairs and the corresponding resolutions. Averaging 5 pairs will give you better resolution inherently. It would seem that Fig 4 would make a lot more sense if you showed it in gradient units, since the central claim of your paper is that you have made a gradiometer. Perhaps you should make a Fig 4b that shows this. 

In Fig 4, the theory begins to deviate from experiment the farther away from the source that you are. Why do you think this is?

You claim that your sensitivity is comparable or better than MEMS, Hall effect, Fluxgate, and MR in line 131-133. This is not true. Please change this to something closer to, "the sensitivity of the EPHEG is in the range of high resolution magnetic gradiometry techniques, such as MEMS, Hall effect, Fluxgate, and magnetoresistive." Your sensor footprint is comparable to MEMS and MR sensors, and it may have manufacturing or cost advantages that are not yet known, which makes this valuable. Additionally, your ability to measure at different positions so close to each other is extremely valuable and is likely not possible in these other systems. Such a feature can be used to enhance accuracy for very low field measurements, as you have shown. Or they may be used for extremely high spatial precision for larger fields. Additionally, as you mentioned, the ability to cancel out common fields is greatly enhances by the ability to put these leads close together. You may even be able to cancel out other types of interference such as temperature. 

There are several grammatical errors that need to be cleaned up. In line 30, there is a run-on sentence after portability that should be split into two sentences. In line 36, "have" should be "has." In line 46, "is" should be "are. In line 80, "EMNs closely resemble our previous devices with similar thickness..." Please correct these and many others that I did not point out. 
